DISCOVERY REPORT

# Humans and great apes visually track event roles in similar ways

**Vanessa A. D. Wilson**[1,2,3]*, **Sebastian Sauppe**[3,4,5], **Sarah Brocard**[1], **Erik Ringen**[2,3,6], **Moritz M. Daum**[3,4,5], **Stephanie Wermelinger**[4,5], **Nianlong Gu**[3,6], **Caroline Andrews**[2,3], **Arrate Isasi-Isasmendi**[2,3], **Balthasar Bickel**[2,3‡], **Klaus Zuberbühler**[1,3‡]

**1** Department of Comparative Cognition, Institute of Biology; University of Neuchatel, Neuchatel, Switzerland, **2** Department of Comparative Language Science, University of Zurich, Zurich, Switzerland, **3** Center for the Interdisciplinary Study of Language Evolution, University of Zurich, Zurich, Switzerland, **4** Department of Psychology, University of Zurich, Zurich, Switzerland, **5** Jacobs Center for Productive Youth Development, University of Zurich, Zurich, Switzerland, **6** NCCR@LiRI Group, Linguistic Research Infrastructure, University of Zurich, Zurich, Switzerland

‡ These authors are joint senior authors on this work.
* Vanessa.Wilson@hull.ac.uk

The Editors encourage authors to publish research updates to this article type. Please follow the link in the citation below to view any related articles.

**Data Availability Statement:** Data and code are available at https://osf.io/47wap/?view_only=

## Abstract

Human language relies on a rich cognitive machinery, partially shared with other animals. One key mechanism, however, decomposing events into causally linked agent–patient roles, has remained elusive with no known animal equivalent. In humans, agent–patient relations in event cognition drive how languages are processed neurally and expressions structured syntactically. We compared visual event tracking between humans and great apes, using stimuli that would elicit causal processing in humans. After accounting for attention to background information, we found similar gaze patterns to agent–patient relations in all species, mostly alternating attention to agents and patients, presumably in order to learn the nature of the event, and occasionally privileging agents under specific conditions. Six-month-old infants, in contrast, did not follow agent–patient relations and attended mostly to background information. These findings raise the possibility that event role tracking, a cognitive foundation of syntax, has evolved long before language but requires time and experience to become ontogenetically available.

## Introduction

Language is considered unique to humans, a distinction which leads to the prevailing question of how it has evolved. An empirical strategy has been to identify the cognitive mechanisms that language relies on and to reconstruct their evolutionary history using comparative research with humans and other animals. One important cognitive mechanism is the propensity for speakers and listeners to decompose events into causally structured agent–patient relations [1]. For example, a sentence like "Alice picked up the caterpillar" has Alice as the agent and the caterpillar as the patient. This distinction is deeply entrenched in meaning, neuroanatomically detectable [2] and responsible for core syntactic phenomena, such as case marking or constituent hierarchies [3], with only very few exceptions across the world's languages [4].

8c2b20667fc441178269291fda5262bf. Interactive figures are also available at https://dataplatform.evolvinglanguage.ch/eventcog-eyetracking/.

**Funding:** The National Center for Competence in Research "Evolving Language" (SNSF agreement number 51NF40_180888, B.B., K.Z., M.M.D.): https://evolvinglanguage.ch/ Swiss National Science Foundation (project grant numbers 310030_185324, K.Z): https://www.snf.ch/en Swiss National Science Foundation (100015_182845, B.B.): https://www.snf.ch/en The National Center for Competence in Research "Evolving Language" Top-Up Grant (grant number N603-18-01, V.A.D.W., K.Z., B.B., M.M.D.): https://evolvinglanguage.ch/ Foundation for Research in Science and the Humanities at the University of Zurich (grant number 20-014, V.A.D.W., S.S., B.B.): https://www.research.uzh.ch/en/funding/researchers/stwf.html Seed money grant, University Research Priority Program "Evolution in Action", University of Zurich (S.S.): https://www.evolution.uzh.ch/en.html Jacobs Foundation (S.W., M.M.D.): https://jacobsfoundation.org/ Swiss National Science Foundation (grant number PZ00P1_208915, S.S.): https://www.snf.ch/en The funders had no role in study design, data collection and analysis, decision to publish, or preparation of the manuscript.

**Competing interests:** The authors have declared that no competing interests exist.

Furthermore, languages privilege agent over patient roles, preferring the simplest and least specific marking for them (e.g., nominative case) [5], even though this often incurs ambiguity and additional neural activity during sentence planning [6,7]. Correspondingly, agents tend to be named before patients [8,9], a trend only matched in sentence structure by a concurrent preference for placing reference to humans before reference to inanimate things [10].

These biases in linguistic expression build on resilient mechanisms in human event cognition [11]. For example, when apprehending the gist of events from still pictures, people tend to be quicker to identify agents than patients [12] and assign agency almost instantly and with remarkably little variation across cultures and languages [13]. Early agent identification is typically followed by distributed attention between agents and patients in a processing stage known as "relational encoding" during early sentence planning [6]. The same resilience is also apparent in comprehension during sentence processing. When sentences violate expectations about agent and patient roles, for example, when it turns out that a noun referred to a patient instead of an agent, neurophysiological measures indicate that agency is usually assigned as the initial default, even when this goes against usage probabilities and the rules of grammar. This is evidenced by a negativity in event-related potentials (the N400 effect) when ambiguous sentences are resolved towards patient-before-agent order, violating comprehenders' expectations about the primacy of agents, which accordingly should be expressed first (similar to: The girl . . . was chased by the dog) [5,14–16]. Together, these findings suggest that language builds on a universal neurocognitive mechanism of event decomposition to make sense of the world and its linguistic representations.

This raises the question of how human event cognition has evolved. We are not aware of any evidence in the animal communication literature that demonstrates that signals can refer to agent–patient interactions, neither in natural communication nor with artificial languages [17]. One hypothesis, therefore, is that nonhuman animals (from here on, "animals") do not possess the cognitive resources for decomposing events into agents and patients. Certainly, animals can comprehend aspects of physical causality (e.g., that pushing causes falling) [18]. Still, it is unclear whether this is due to perceptions of simple co-occurrences or more complex perceptions of events as agent activities causing patient changes. Related to this, although there is little doubt that animals perceive the participants of events, their attention may be absorbed by the participants' social attributes, such as their identities, social roles [19], or behavioral intentions [18,20], all of which predict large situational and individual variation in how events are processed.

The alternative hypothesis, to be tested here, is that animals are capable of human-like event decomposition [1], but do not have the motivation or the resources to communicate about agent–patient relations. To explore this, we tested how participants across closely related species of hominids perceived a range of naturalistic events that would elicit causal processing in humans. We compared gaze responses to short video clips between members of the 4 genera of great apes—humans (*Homo sapiens*), chimpanzees (*Pan troglodytes*), gorillas (*Gorilla gorilla*), and orangutans (*Pongo abelii*) (see S1 Fig for ape setup). We also tested human infants at 6 months old, before they start to actively use language and while still developing linguistic processing abilities. By this age, infants already show an impressive cognitive toolkit: they are sensitive to goal-directed actions and agency [21], track changes in goal-directed behavior [22], and extract key information from video stimuli to understand events [23]. At the same time, young infants struggle to process goal predictions [24], and neural selectivity of third-party social interactions does not begin to emerge until after 9 months [25]. As infants develop, language and event perception become increasingly intertwined, as documented by the way verbs and actions are processed [26].

Currently, our understanding of agent-privileged event cognition in humans rests mainly on paradigms that use static stimuli, which are often artificial or overly simplistic and do not reflect the complexity of real-life interactions. In this study, we used dynamic scenarios across a broad range of natural events to compare overt visual attention to agents and patients as the actions unfold. Scenarios were presented as $N$ = 84 short (2 to 10 s long), silent color video clips, depicting animate agents and both animate and inanimate patients of (unfamiliar) humans, chimpanzees, gorillas, and orangutans engaged in natural interactions. This included scenes such as grooming, play, eating and object manipulation among apes (both in zoos and the wild), and helpful and mildly agonistic interactions, as well as object manipulation, among humans; the latter were all filmed in the same setting with a primarily white backdrop. Further details can be found in S2 Table and in video examples on the OSF repository (https://osf.io/47wap/?view_only=8c2b20667fc441178269291fda5262bf). All participants saw the same stimuli. When creating the videos, we deliberately avoided rigidly controlling for low-level perceptual features, as this would have created sterile footage with low socio-ecological validity [27] and, critically, reduced interest for ape participants. Instead, we presented scenes that sought to capture much greater variation of real life. Possible confounding factors, such as differences in the amount of agent motion or relatively larger sizes of agents or patients between videos, were accounted for in the statistical models (see S1 Methods and S5 and S6 Figs).

We predicted that event roles would be necessary to interpret the event scenarios—that is, the distinction between agent and patient is needed to explain gaze distribution to the individuals depicted. For human adults, we expected to see early and overall agent biases, consistent with previous findings, but with attention patterns mediated by the progression of the action rather than the need to extract agent–patient information rapidly, as in brief exposure studies [13]. We predicted that if event decomposition were a general feature of great ape cognition and present without language, then visual attention should not differ across the 4 species. In particular, we predicted earlier attendance to agents than patients, in line with the privileged status of agents in language and gist apprehension in still pictures. Alternatively, if event decomposition were uniquely human—or dependent on language—we expected to find this pattern only in adult humans, and large and random variation in the nonhuman primates, which would likely depend on low-level features such as color or contrast of the stimuli. Regarding the infants, if event-role decomposition required experience gained through observing third-party interactions, we expected to see differences in how they attended to agents and patients compared with adults.

## Results

For the analysis, we first fitted Dirichlet models to examine whether event role distinctions better explained gaze patterns than any of the covariates (such as relative size of the individuals depicted). The model included a number of predictor variables, including the size of our areas of interest (AOI: agents, patients and other information) and the difference in movement between agents and patients, as well as participant ID and participating species, trial ID, species depicted, and time (see Statistical analyses in the S1 Methods for more details). We compared one model that included the distinction as a predictor of gaze proportions over each trial with a model that did not include the role distinction among the predictors, treating them the same way as when 2 individuals did not interact with each other (see Trial averaged model comparison in the S1 Methods). Results indicated that the role distinction substantially improved model fit: all models with the role distinction leverage over 90% of the weight in data prediction under leave-one-out cross-validation (S1 Table), indicating that participants did not simply differentiate between 2 individuals, but between the roles attached to them.

Given the evidence for the role distinction, we next conducted a time series analysis to examine the probability of gaze to the agent, patient, or other areas over 5% time intervals (see Time series model in the S1 Methods). The purpose of this model was to determine how attention to agents and patients varied and developed as the events unfolded. We distinguished scenes depicting social interactions (i.e., with animate patients) and scenes depicting nonsocial interactions (i.e., with inanimate patients). The model included the same predictor variables as the Dirichlet model (see Statistical analyses in the S1 Methods for more details).

Preliminary examination of the data revealed large heterogeneity in the time course data between videos, which was not captured by either differences in inanimate and social videos, or relative AOI size (see S1 Methods for more details on accounting for differences in AOI size). To determine where these gaze differences came from, we explored the videos with heterogeneous responses, which concerned specifically inanimate videos of apes. From these scenarios, we identified 4 categories that might explain the resulting variance: (1) scenes generally containing food; (2) tool use, such as nut cracking or honey dipping; (3) instances where the agent or patient were oriented towards the camera; and (4) instances where the agent or patient had direct gaze towards the camera, something that could have been perceived as threatening by some ape participants (for details, see S1 Methods and S3 Table). To estimate the effects of these categories, we fitted additional models (figures for these models can be viewed in a dashboard at https://dataplatform.evolvinglanguage.ch/eventcog-eyetracking/). These models indicated a clear difference in gaze patterns to inanimate scenes containing food, compared to those containing objects. We thus further split the inanimate videos into those containing food or not.

Generalized additive models, such as ours, are best understood by examining model predictions given different values of the covariates (time, species, condition), rather than interpreting individual coefficients. This is due to both the large number of parameters and the nonlinearity induced by the link function. As such, we draw on the credible intervals of the models' predictions at multiple time points to interpret the data [28] (see, for example, Figs 1 and 2).

Five main findings emerged. Firstly, in both adult humans and apes, after accounting for attention to background information, there was an early focus on the action with attentional switches between agents and patients, suggesting that participants engaged in relational encoding [6], similar to what humans do when asked to describe actions depicted in still images (Fig 1 and on the OSF repository (S4 Table, https://osf.io/47wap/?view_only= 8c2b20667fc441178269291fda5262bf)). Secondly, in both adult humans and apes, the inanimate food condition triggered a striking bias towards agents at the onset of the action (with posterior probabilities for log odds ratios of agent versus patient fixations all being above 0), absent from the other conditions (Fig 1). For humans, agents remained salient throughout the scenario, while for apes, this gaze pattern was less pronounced. Thirdly, the data revealed an initial peak in agent looks in the inanimate scenarios compared with social scenarios. This suggested that animate agents acting on inanimate patients were easier to identify as agents, compared with a scenario where both agent and patient were animate. This indicated that the type of patient in a scenario could initiate different expectations about agency, meaning that participants clearly differentiated agents and patients as distinct entities from the start of the video scenes, further bolstering our findings from the Dirichlet model about the importance of assigning event roles for scene interpretation. Fourthly, human adults directed most of their visual attention to agents and patients, while apes attended more to other information. This effect was slightly stronger for orangutans (Figs 2 and on the OSF repository (S4–S6 Tables, https://osf.io/47wap/?view_only=8c2b20667fc441178269291fda5262bf)), suggesting a possible phylogenetic emergence of event role attribution. Given however that orangutans were eye tracked through the mesh rather than through plexiglass, it is also possible that this difference

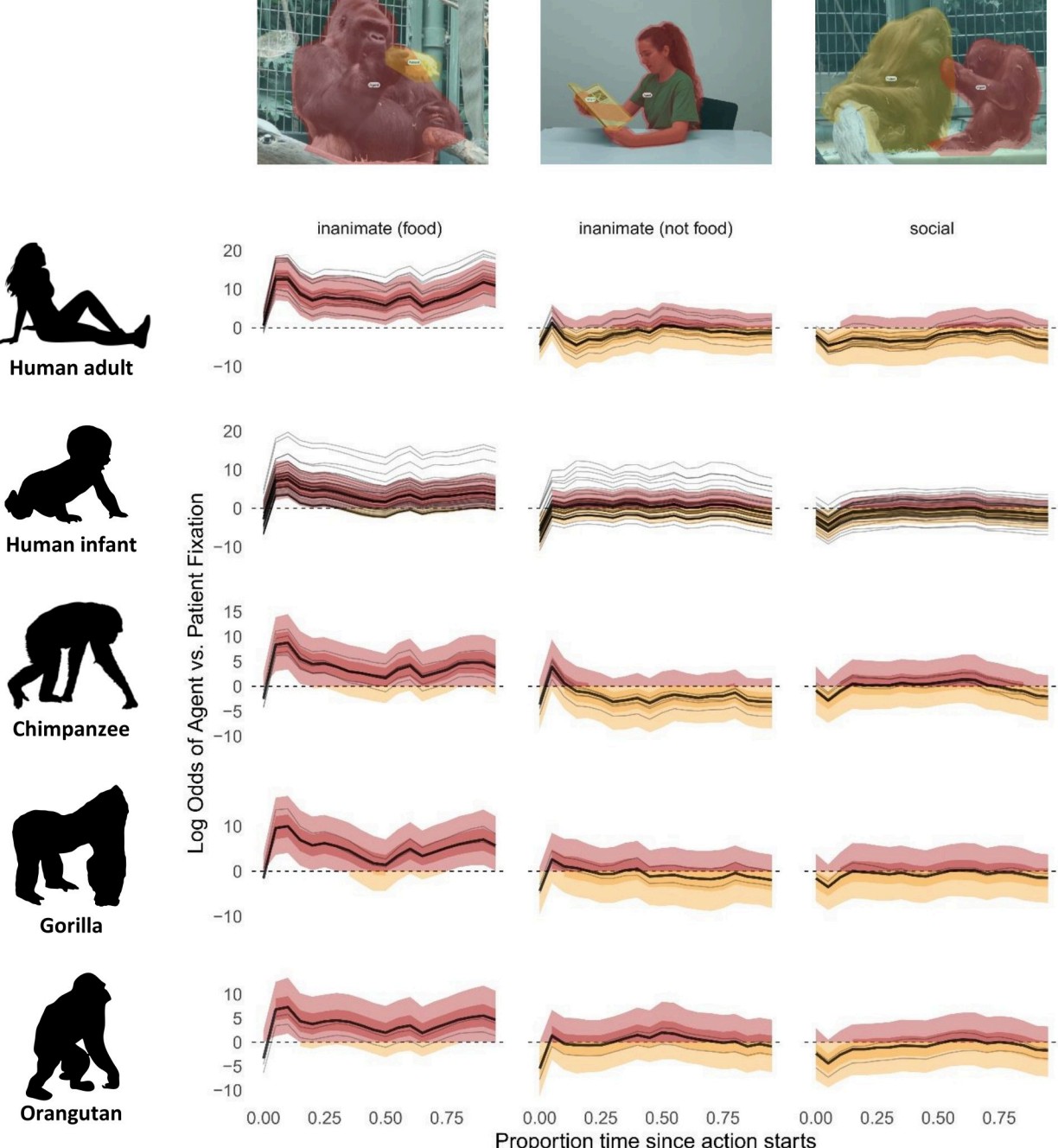

**Fig 1.** Model predictions of log odds ratio for fixation to either agent (red) or patient (orange) over time. Thick lines represent the grand mean, thin lines represent individual participants. Time point 0 on the x-axis indicates action start time, normalized across stimuli. Light shaded ribbons indicate 90% credible intervals. When these exceed 0, there is a 90% probability of gaze to the agent (light red); when they are below 0, there is a 90% probability of gaze to the patient (light orange); when they include zero, there is a 90% probability that gaze alternates between agent and patient. The darker shaded ribbons indicate the middle 50% of the posterior probability mass.

is attributable to poorer gaze detection (see Apparatus and Procedure in S1 Methods). It is also necessary to consider that, due to small sample size, the effect was due to noise in the data. Finally, human infants significantly differed from the other groups, by attending mostly to

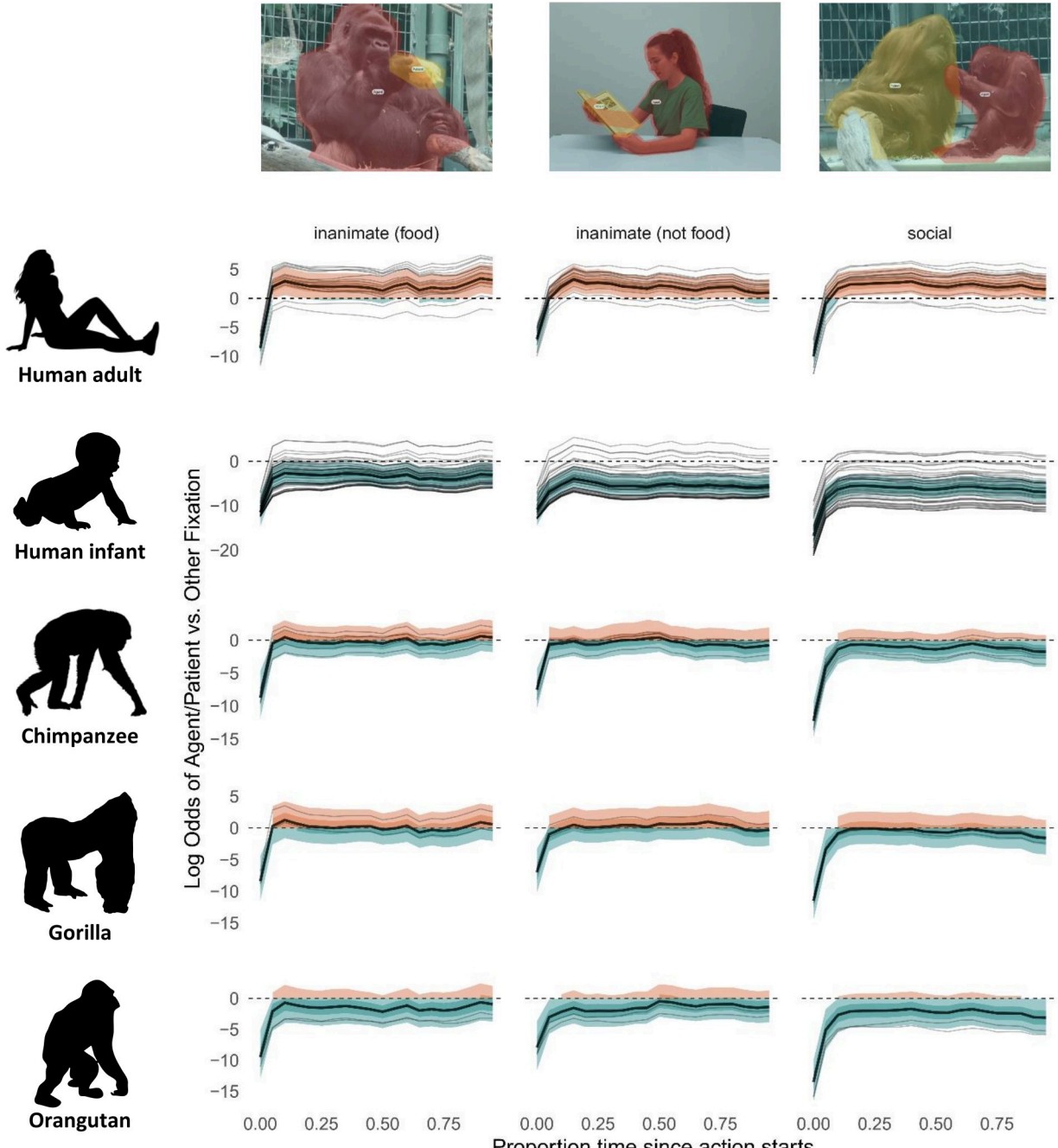

**Fig 2.** Model predictions of log odds ratio for fixation to agent or patient (orange) versus other (turquoise) over time. Thick lines represent the grand mean, thin lines represent individual participants. Time point 0 on the x-axis indicates action start time, normalized across stimuli. Light shaded ribbons refer to 90% credible intervals. They exceed 0 for humans, indicating gaze priority for agents and patients (orange). They are below 0 for human infants (turquoise) (with the exception of inanimate food), indicating gaze primarily to background information; 90% credible intervals for apes border around 0, indicating that attention is divided between agents and patients (orange) and other information (turquoise). Darker shaded intervals indicate the middle 50% of the posterior probability mass.

other, non-agent and non-patient, information within each scene (Figs 2 and S7). To explore the data, we have provided a tool on the OSF repository (a Shiny app dashboard) and via a url at https://dataplatform.evolvinglanguage.ch/eventcog-eyetracking/ that allows readers to

visualize the effects of different variables on gaze patterns. See also S8 and S9 Figs for time-aggregated fixations to AOIs and stimulus per species.

## Discussion

Evolutionary theories of syntax have focused mainly on how formal complexity has emerged [29–31], whereas the underlying cognitive mechanisms have rarely been addressed. Here, we tested a cognitive hypothesis, which proposes that central aspects of human syntax, such as case-marking or constituent hierarchy, build on a prelinguistic cognitive mechanism that decomposes events into causally structured agent–patient relations [1]. To test this, we exposed apes to stimuli that elicit causal processing in humans and compared the gaze patterns between humans and nonhuman apes. Participants across species tracked events in strikingly similar ways, focusing on the action between agents and patients in a manner reminiscent of relational encoding for planning to speak [6]. This finding suggests that apes, like human adults, decompose the causal agent–patient roles depicted. The only noticeable difference was that nonhuman apes showed more visual exploration of background information than human adults, perhaps due to differences in experience with watching and interpreting videos or higher intrinsic interest in scanning the larger environment. This is reflected in S2 and S3 Figs, where human adult attention to agents and patients is more pronounced, because these results do not account for attention to "other" information (S4 Fig). Notably, apes' looking behavior showed more similarity to human adults than did human infants. If apes were unable to track agent–patient relations, we would expect attention patterns similar to those seen in infants. These observations are compatible with the interpretation that event decomposition did not emerge as a unique form of human cognition together with language. Rather, our findings comprise another piece of evidence suggesting a cognitive continuum between humans and nonhuman apes, albeit in a novel cognitive domain.

Unexpectedly, across all event categories, in neither humans nor great apes did we find a strong bias towards agents (with the exception of food scenarios). This is in contrast to findings from a large body of previous research using static images, as well as a recent comparative study examining event role preferences [32]. This difference is likely due to the nature of the task. Previous studies asked for rapid decisions between roles [13,32], similar to when listeners have to come up with on-the-fly predictions on roles while processing a sentence [33,34]. It is likely that an agent bias manifests itself primarily under these high-demand conditions, while it is not as relevant when watching an event that unfolds over time [35]. Additionally, unlike some previous studies [12,36] although see [13], we controlled for size and movement of the areas of interest, as well as event type; size of the agent or patient, in particular, has a strong effect on gaze probability (S5 Fig). Curiously, when considering gaze differences between event categories, we found the strongest agent bias in video scenes depicting interactions with food. One possible explanation is that, in social species, attending to agents who have access to food provides a survival advantage. This points to social learning as a possible precursor for semantic role attribution. An intriguing possibility requiring further studies, therefore, is that the agent bias reported elsewhere has its roots in trophic interactions. A more parsimonious explanation could be that agency bias is strong in these scenes because food items are less interesting as patients compared to objects. Notably, since size of agents and patients was accounted for in the models, size cannot explain the gaze bias that we found towards agents. Further research is needed to differentiate these different possibilities and to more systematically explore different degrees and kinds of cognitive pressure in event cognition and different ecological contexts of events.

In contrast to human adults, the 6-month-old infants attended to agents and patients with very low probability. One explanation may be that while infants this young show sophisticated perception and interpretation skills of causality in social interactions in visually and cognitively uncomplicated material [37,38] and with previous familiarization or habituation (i.e., learning) phases, they may still have trouble detecting causality in complex visual material [39], especially in social interactions and with no prior controlled learning phase. Indeed, our stimuli were more complex than typically used in infant studies. Also, they were not directed towards the infants and did not include any communicative information towards the observing infant [38]. However, these were chosen to reflect real-life diversity of events. Key ingredients, such as event parsing [40], causal integration across scenes [24], and triadic awareness [41], are known to develop gradually during the first 12 months, suggesting that our content was too challenging and probably too alien to their existing world experience. Another plausible hypothesis is that processing of dynamic natural scenes requires computational resources and oculomotor control not yet sufficiently developed at this age [42]. As a consequence, integration and interpretation of relevant information is more time-consuming [22] or just not yet possible. Given that event categorization relies on neurally constructed models, which are updated with experience [43], it is likely that, at this age, infants are still developing the event models that will guide their attention to scene information.

In sum, our study demonstrates that nonhuman great apes and human adults show similar looking behavior towards agent–patient interactions, consistent with the notion of a shared underlying cognitive mechanism. The fact that infants show a significantly different looking pattern than both human adults and apes suggests that proficiency in language is not driving the observed looking pattern. Earlier explanations that relied on human–animal morphological differences in vocal tracts [44], lack of declarative communication [45], or lack of call composition [46] no longer stand. Our results add to the shared cognitive foundations of language by suggesting that event decomposition, a foundation of syntax, evolved before language, on par with signal combinations [47], theory of mind [48], and joint commitment [49].

What has happened then during human evolutionary history that allowed us to map event roles into verbal expressions? We can think of three probably interlinked evolutionary transitions that may have paved the way from primate-like communication to human language: (a) changes in social cognition; (b) changes in communicative needs; and (c) changes in expressive power. Regarding social cognition, the key step may have been to externalize event cognition through language, by moving from implicit to explicit attributions. For example, compared to chimpanzees, adult humans attend more to an agent's face following an unexpected action outcome, as if explicitly trying to discern the actor's mental state [50]. Exploring how apes attend to detailed social scenarios could help to understand the differences that led to this next step in humans. Regarding communicative needs, one argument is that increased levels of cooperation and coordination brought about increased communicative needs, a convergent evolution independent of wider cognitive evolution [51,52]. Comparative studies that delve deeper into the emergence of cooperative behavior could help to further elucidate this hypothesis [53]. Finally, regarding expressive power, modern humans roughly have 3-fold larger brains than chimpanzees with vastly more computational power, allowing for processing of more varied signal structures. Although there are examples of limited compositionality in animal communication, there is no evidence for free variation and creative use [47]. The study of ape communication, however, continues to reveal new insights, which should seek to confirm the degree of difference in flexible communication compared to humans. Testing these hypotheses may provide further answers in the quest for the origins of language by better understanding why nonhuman apes do not communicate in the same way as humans do, despite an increasingly closing gap with human cognitive abilities.

## Methods

Study methods are described in full in the Supporting information (S1 Methods).

### Ethics statement

Ethical approval for the ape research was provided by the Canton of Basel Veterinary Office (approval numbers 2983 and 3077) and by the Animal Welfare Officer at Basel Zoo. All apes participated voluntarily. They were not separated from their group during testing, nor were they food or water deprived, and could leave at any time. They were rewarded for participation with diluted sugar-free syrup, provided in restricted quantities as approved by the zoo's veterinary team.

Ethical approval for the human research was approved by the local ethics committee of the University of Zurich (approval numbers 18.10.9 and 21.9.18) and performed in accordance with the ethical standards of the 1964 Helsinki Declaration and its later amendments. All adult participants or the infants' caregivers gave informed written consent before data collection. After completing the task, adult participants received CHF 20 and infant participants were rewarded with a certificate and a small present worth approximately CHF 5.

## Supporting information

**S1 Methods. Supplementary materials.**
(DOCX)

**S1 Fig. Eye-tracking setups at Basel Zoo.**
(DOCX)

**S2 Fig. Time course of gazes towards agent in the videos, predicted with a categorical Bayesian multilevel model.**
(DOCX)

**S3 Fig. Time course of gazes towards patient in the videos, predicted with a categorical Bayesian multilevel model.**
(DOCX)

**S4 Fig. Time course of gazes towards other information in the videos, predicted with a categorical Bayesian multilevel model.**
(DOCX)

**S5 Fig. AOI fixation probability as a function of AOI size.**
(DOCX)

**S6 Fig. AOI fixation probability by movement difference between agent and patient.**
(DOCX)

**S7 Fig. Gaze for infant participants when viewing human-only footage (top panel) and ape-only footage (bottom panel) to agent (left), patient (middle), and other (right).**
(DOCX)

**S8 Fig. Proportion of fixations to each area of interest, aggregated across time points.**
(DOCX)

**S9 Fig. Average proportion of fixations for each species x stimulus, averaging over time, trial, and participant.**
(DOCX)

**S1 Table. Model comparison differentiating event role specification as a predictor of agent–patient gaze proportion.**
(DOCX)

**S2 Table. Full list of video footage presented.**
(DOCX)

**S3 Table. Variables coded from video stimuli.**
(DOCX)

## Acknowledgments

For support and assistance at Basel Zoo, we thank Adrian Baumeyer, Fabia Wyss, Raphaela Heesen, Stephan Lopez, Gaby Rindlisbacher, Rene Buob, Roland Kleger, Jonas Schaub, Nicole Fischer, Reto Lehmann, Corinne Zollinger, Markus Beutler, Dominic Hohler, Patrick Wyser, Flurin Baer, Amanda Spillmann, Stephan Argast, and the technician team. We also thank Carla Pascual for help with data collection. For providing footage from apes, we thank Emily Genty, Cat Hobaiter, Jennifer Botting, Erin Stromberg, and Atlanta Zoo, as well as Zurich Zoo for allowing us to film their apes. We thank Sara I. Fabrikant and Tumasch Reichenbacher for providing their eye-tracking laboratory for the human adult data collection and Sina Nägelin, Nina Philipp, and Deborah Lamm for collecting the data from human adults and infants, as well as Marco Bleiker for technical assistance with infant data collection. We thank Sebastien Quigley and Carla Pascual for assistance in data processing. We also thank Shreejata Gupta, Christopher Krupenye, and Josep Call for their advice on establishing the eye-tracking setups for data collection from great apes.

## Author Contributions

**Conceptualization:** Vanessa A. D. Wilson, Sebastian Sauppe, Sarah Brocard, Moritz M. Daum, Stephanie Wermelinger, Balthasar Bickel, Klaus Zuberbühler.

**Data curation:** Vanessa A. D. Wilson, Sarah Brocard, Erik Ringen, Stephanie Wermelinger, Nianlong Gu.

**Formal analysis:** Vanessa A. D. Wilson, Sebastian Sauppe, Erik Ringen, Balthasar Bickel, Klaus Zuberbühler.

**Funding acquisition:** Vanessa A. D. Wilson, Sebastian Sauppe, Moritz M. Daum, Balthasar Bickel, Klaus Zuberbühler.

**Investigation:** Vanessa A. D. Wilson, Sarah Brocard.

**Methodology:** Vanessa A. D. Wilson, Sebastian Sauppe, Sarah Brocard, Moritz M. Daum, Stephanie Wermelinger, Balthasar Bickel, Klaus Zuberbühler.

**Project administration:** Vanessa A. D. Wilson.

**Resources:** Vanessa A. D. Wilson, Sebastian Sauppe, Sarah Brocard, Moritz M. Daum, Caroline Andrews, Arrate Isasi-Isasmendi, Balthasar Bickel, Klaus Zuberbühler.

**Software:** Sebastian Sauppe, Erik Ringen, Nianlong Gu.

**Supervision:** Balthasar Bickel, Klaus Zuberbühler.

**Validation:** Erik Ringen.

**Visualization:** Vanessa A. D. Wilson, Sebastian Sauppe, Erik Ringen, Balthasar Bickel.

**Writing – original draft:** Vanessa A. D. Wilson.

**Writing – review & editing:** Vanessa A. D. Wilson, Sebastian Sauppe, Sarah Brocard, Erik Ringen, Moritz M. Daum, Stephanie Wermelinger, Nianlong Gu, Caroline Andrews, Arrate Isasi-Isasmendi, Balthasar Bickel, Klaus Zuberbühler.

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
