## [Editor Report · Decision Letter 0]

14 May 2024

Dear Dr Wilson, 

Thank you for submitting your manuscript entitled "Primate origins of human event cognition" for consideration as a Research Article by PLOS Biology and apologies for the long delay in getting back to you. Unfortunately, we had difficulties getting advice from the Academic Editor who handled the original submission, so we have discussed your manuscript within the PLOS Biology editorial team and I am writing to let you know that we would like to send your submission out for external peer review.

Please note, however, that we are not convinced that the reviewers' concerns are fully addressed in your revised manuscript and don't think that the requested additional evidence is provided. However, as the reviewers were quite interested in your study, we think your manuscript may be suitable for our Discovery Reports format (https://journals.plos.org/plosbiology/s/what-we-publish#loc-discovery-report). You don't need to fully reformat your manuscript at this point to fit into this format (as we don't know whether the reviewers will be satisfied with the revision even for this format), but we would like to encourage you to select "Discovery Report" when completing your submission as described below. We will then send the manuscript back to the reviewers, asking them to review your study as a Discovery Report. 

Once your full submission is complete, your paper will undergo a series of checks in preparation for peer review. After your manuscript has passed the checks it will be sent out for review. To provide the metadata for your submission, please Login to Editorial Manager (https://www.editorialmanager.com/pbiology) within two working days, i.e. by May 16 2024 11:59PM.

Kind regards,

Christian

Christian Schnell, PhD

Senior Editor

PLOS Biology

cschnell@plos.org

---

## [Decision Letter · Decision Letter 1]

11 Jul 2024

Dear Dr Wilson,

Thank you for your patience while we considered your revised manuscript "Primate origins of human event cognition" for consideration as a Discovery Report at PLOS Biology. Your revised study has now been evaluated by the PLOS Biology editors, the Academic Editor and two of the original reviewers. Please allow me first of all to apologize for the long delay in sending our decision. Unfortunately, the previous academic editor was no longer available to handle your manuscript and it has taken some time until I was able to find a new academic editor. 

In any case, in light of the reviews, which you will find at the end of this email, we are pleased to offer you the opportunity to address the remaining points from the reviewers in a revision that we anticipate should not take you very long. We will then assess your revised manuscript and your response to the reviewers' comments with our Academic Editor aiming to avoid further rounds of peer-review, although might need to consult with the reviewers, depending on the nature of the revisions.

**IMPORTANT - SUBMITTING YOUR REVISION**

*Resubmission Checklist*

*Published Peer Review*

*PLOS Data Policy*

*Blot and Gel Data Policy*

Sincerely,

Christian

Christian Schnell, PhD

Senior Editor

PLOS Biology

cschnell@plos.org

REVIEWS:

Reviewer #1: I think the revised manuscript is more clearly written. I also appreciated their justification and further explanation of their statistical analyses. I still felt like the paper would benefit from some more straightforward analyses to give the reader a better intuition and more confidence in the results.

I understand now that the Dirichlet model provides a measure of time looking at each AOI. I still think the percent of time looking at each AOI (calculated simply as the time at AOI/total time) without the full model would be informative. I don't insist the authors add this to the paper if the results look similar to the existing analyses, but I would be curious to see the results and this seems very simple to calculate.

I also still think that comparing heatmaps across species is important. I think the authors misunderstood my initial request - I would like to see a single heatmap per video (not time point) to quantify overall looking similarity. This would lose temporal resolution, but I believe would be a nice complement to the current time series analyses.

I thought the videos on OSF were somewhat confusing. In many cases, there seemed to be some panels that were not clearly labeled. E.g., video inanimate_human_compilation the bottom, left panel seems to be the infant looking behavior, but it is unclear what the other panels are.

Further, it seems from most of these example videos that babies are indeed looking at agents in the videos, perhaps it was just these examples, but this underscores my desire to see the above heatmap or AOI analyses.

I appreciated the authors efforts to make the data publicly accessible, but the need to install third party software was a barrier to me exploring the data.

Pg 3, line 205, reference 25 (Farris 2022) seems to be incorrectly summarized, as the main finding is neural sensitivity to observed social interactions.

Line 754 there is a typo : Trail > Trial

Reviewer #2: I appreciate the efforts that the authors have made in the revision process. Overall, the manuscript is now much clearer, and much more appropriate for publication, though I still have a few more minor recommendations:

There are a few places where the authors could still temper the language slightly, given that the referenced touchscreen data that bolster the agent-patient decomposition claim are not part of this manuscript:

-In line 40 (abstract), personally I feel that it would be more appropriate to say "These findings raise the possibility that event role tracking, a cognitive foundation of syntax, evolved long before language but requires time and experience to become ontogenetically available."

-Line 319 change 'indicating' to 'suggesting'

Given the graph produced in response to reviewer 2's concerns, I think it would be appropriate for the authors to mention that agent-patient size differences could be a strong driver of the agent bias found only in the food scenes.

I also agree with reviewer 1 that additional analyses on the infant data, restricted to trials in which infants were watching human videos only, would be informative. It would help clarify whether infants show a general lack of 'agent-patient viewing patterns' to complex/dynamic scenes or whether these patterns are manifest when the stimuli are more familiar. That said, the presence or lack thereof of this additional analysis does not disrupt the authors' primary findings or claims about similarities between humans and nonhuman apes.

---

## [Editor Report · Decision Letter 2]

29 Aug 2024

Dear Dr Wilson,

Thank you for your patience while we considered your revised manuscript "Primate origins of human event cognition" for publication as a Discovery Report at PLOS Biology. This revised version of your manuscript has been evaluated by the PLOS Biology editors and the Academic Editor.

Based on our Academic Editor's assessment of your revision, we are likely to accept this manuscript for publication, provided you satisfactorily address the remaining points raised by the Academic Editor. Please also make sure to address the following data and other policy-related requests:

* We would like to suggest a different title to improve accessibility: "Humans and great apes show similar capabilities of event role tracking in a comparative eye tracking study"

* Please add the links to the funding agencies in the Financial Disclosure statement in the manuscript details

* In your ethics statement, please state whether the participants or their caregivers provided written or oral consent.

* DATA POLICY:

Regardless of the method selected, please ensure that you provide the individual numerical values that underlie the summary data displayed in the following figure panels as they are essential for readers to assess your analysis and to reproduce it: S6 and S8

CODE POLICY

We expect to receive your revised manuscript within two weeks. 

*Published Peer Review History*

*Press*

Sincerely,

Christian

Christian Schnell, PhD

Senior Editor

cschnell@plos.org

PLOS Biology

Academic Editor remarks:

1. It does not seem like the two new supplemental figures are referenced in the main text.

2. Line 104: "neural distinction"  "neural selectivity"

---

## [Editor Report · Decision Letter 3]

20 Sep 2024

Dear Dr Wilson,

Thank you for the submission of your revised Discovery Report "Humans and great apes visually track event roles in similar ways" for publication in PLOS Biology. On behalf of my colleagues and the Academic Editor, Leyla Isik, I am pleased to say that we can in principle accept your manuscript for publication, provided you address any remaining formatting and reporting issues. These will be detailed in an email you should receive within 2-3 business days from our colleagues in the journal operations team; no action is required from you until then. Please note that we will not be able to formally accept your manuscript and schedule it for publication until you have completed any requested changes.

When you attend to the requests to come, please also address two final comments from the Academic Editor:

1. Supplemental figures 7, 8 and 9 need also to be referenced in the main text.

2. Line 104: "neural distinction"  "neural selectivity"

PRESS

Sincerely, 

Christian

Christian Schnell, PhD

Senior Editor

PLOS Biology

cschnell@plos.org